# Telomere Length Shortening in Microglia: Implication for Accelerated Senescence and Neurocognitive Deficits in HIV

**DOI:** 10.3390/vaccines9070721

**Published:** 2021-07-01

**Authors:** Chiu-Bin Hsiao, Harneet Bedi, Raquel Gomez, Ayesha Khan, Taylor Meciszewski, Ravikumar Aalinkeel, Ting Chean Khoo, Anna V. Sharikova, Alexander Khmaladze, Supriya D. Mahajan

**Affiliations:** 1Medicine Institute, School of Medicine, Infectious Diseases, Drexel University, Positive Health Clinic, Allegheny General Hospital, Allegheny Health Network, Pittsburgh, PA 15212, USA; Chiubin.hsiao@ahn.org; 2Department of Medicine, Division of Allergy, Immunology & Rheumatology, University at Buffalo’s Clinical Translational Research Center, Buffalo, NY 14203, USA; harneetb@buffalo.edu (H.B.); raquel-gomez@uiowa.edu (R.G.); akhan26@buffalo.edu (A.K.); meciszet@my.canisius.edu (T.M.); ra5@buffalo.edu (R.A.); 3Department of Physics, University at Albany SUNY, Albany, NY 12222, USA; tkhoo@albany.edu (T.C.K.); asharikova@albany.edu (A.V.S.); akhmaladze@albany.edu (A.K.)

**Keywords:** HIV-associated neurocognitive disorders (HAND), telomere length, telomerase, microglia, neuro-inflammation, oxidative stress

## Abstract

The widespread use of combination antiretroviral therapy (cART) has led to the accelerated aging of the HIV-infected population, and these patients continue to have a range of mild to moderate HIV-associated neurocognitive disorders (HAND). Infection results in altered mitochondrial function. The HIV-1 viral protein Tat significantly alters mtDNA content and enhances oxidative stress in immune cells. Microglia are the immune cells of the central nervous system (CNS) that exhibit a significant mitotic potential and are thus susceptible to telomere shortening. HIV disrupts the normal interplay between microglia and neurons, thereby inducing neurodegeneration. HIV cART contributes to the inhibition of telomerase activity and premature telomere shortening in activated peripheral blood mononuclear cells (PBMC). However, limited information is available on the effect of cART on telomere length (TL) in microglia. Although it is well established that telomere shortening induces cell senescence and contributes to the development of age-related neuro-pathologies, the effect of HIV-Tat on telomere length in human microglial cells and its potential contribution to HAND are not well understood. It is speculated that in HAND intrinsic molecular mechanisms that control energy production underlie microglia-mediated neuronal injury. TL, telomerase and mtDNA expression were quantified in microglial cells using real time PCR. Cellular energetics were measured using the Seahorse assay. The changes in mitochondrial function were examined by Raman Spectroscopy. We have also examined TL in the PBMC obtained from HIV-1 infected rapid progressors (RP) on cART and those who were cART naïve, and observed a significant decrease in telomere length in RP on cART as compared to RP’s who were cART naïve. We observed a significant decrease in telomerase activity, telomere length and mitochondrial function, and an increase in oxidative stress in human microglial cells treated with HIV Tat. Neurocognitive impairment in HIV disease may in part be due to accelerated neuro-pathogenesis in microglial cells, which is attributable to increased oxidative stress and mitochondrial dysfunction.

## 1. Introduction

HIV cART (combination antiretroviral therapy) has dramatically improved survival rates of HIV infected patients, resulting in an aging HIV infected cohort, and although viral loads remain suppressed in these patients, HIV-associated neurocognitive disorders (HAND) remain highly prevalent and contribute to significant morbidity. Microglial activation occurs in HIV-1 infected subjects, and uncontrolled brain inflammation plays a key role in neuronal injury and cognitive dysfunction during HIV infection. HIV-1 Tat is the transactivator of transcription that is essential for transcriptional regulation and replication of the virus, and is the first protein produced after HIV infection. It stimulates transcription of the HIV-1 genes and regulates gene expression in the host [1,2]. Tat has been implicated in the development of HIV neuro-pathogenesis and is known to be cytotoxic and neurotoxic [3]. HIV-1 Tat protein exerts proinflammatory effects on microglia, astrocytes and neurons that produce key pro-inflammatory cytokines, such as tumor necrosis factor-alpha (TNF-α) and interleukin-1 beta (IL-1β), which ultimately lead to neuron damage and cognitive deficits. Activated microglia also release nitric oxide (NO) and reactive oxygen species (ROS), and contribute to mitochondrial dysfunction, further exacerbating HAND pathogenesis [4,5]. HIV proteins can directly induce neuronal damage. HIV-1 neuropathology results from the neurotoxic effects of the viral proteins and the pro-inflammatory response by microglial cells. HIV-infected microglia respond vigorously to proinflammatory signals and produce an excess of cytokines; increased IL-1β and TNF-α levels are seen in the HAND patients. Immune-activated HIV-infected, brain-infiltrating macrophages and resident microglia release high levels of neurotoxic cytokines, such as TNF-α and IL-1β [6,7,8,9,10].

HIV cART contributes to premature telomere shortening through inhibition of the reverse transcriptase activity of human telomerase by nucleoside reverse transcriptase inhibitors (NRTIs), or induction of oxidative stress and mitochondrial dysfunction [11,12,13]. Adverse effects of cART have been attributed to the mitochondrial toxicity of NRTIs; mitochondrial dysfunction and oxidative stress are also involved in the aging processes. A combination of HIV and cART results in the phenotype of immune senescence, which is found in aging, however here the aging process is accelerated by oxidative stress and mitochondrial dysfunction. The interplay of these events is complex and regulation may occur at a variety of cellular levels, involving genetic and metabolic alterations. Telomere length is a marker of cellular replication capability, and short telomeres have been linked to aging and age-associated diseases. HIV proteins can down-modulate expression of telomerase activity [12,13]. Telomeric DNA loss is cumulative; when telomere length decreases until it reaches a critical length, the cellular surveillance mechanisms are activated and cellular proliferation ceases by permanent cell-cycle arrest or by apoptosis. Telomerase is a cellular reverse transcriptase responsible for the de novo synthesis of telomeric DNA, and requires reverse transcriptase to elongate the telomere. NRTIs, including zidovudine, stavudine, tenofovir, didanosine and abacavir, inhibit telomerase effectively in vitro [11]. Telomerase activity is also inhibited by currently recommended RTI agents, such as tenofovir and abacavir [12,13]. NRTIs can induce oxidative stress, causing mitochondrial dysfunction. Telomeres and telomerase activity have been characterized as biomarkers of cellular aging. Telomeres are coated with a protein complex that consists of a group of proteins such as telomere repeat-binding factor 1 (TRF1) and 2 (TRF2). These prevent the DNA repair machinery from recognizing and processing telomeres during the repair of double-stranded DNA breaks.

We examined the effect of cART on TL and telomerase activity in PBMC obtained from HIV-1 infected patients who had uncontrolled HIV infection and who were on cART and compared them with those who were cART naïve. We observed a significant decrease in telomere length in HIV-1 infected patients who had uncontrolled HIV infection on cART, compared to HIV-1 infected patients who had uncontrolled HIV infection who were cART naïve. Further, we examined changes in oxidative stress, mitochondrial cellular energetics and alterations in expression of telomeric length, telomerase and TRF-1 in human HTHU and the HTHU/HIV microglial cells treated with a combination of tenofovir (TFV) + emtricitabine (FTC) + dolutegravir (DTG) or cART (300 µM TFV + 50 µM FTC + 188 nM DTG) [14], and observed a significant decrease in telomerase activity, telomere length and mitochondrial function, and an increase in oxidative stress in human microglial cells treated with HIV Tat. Microglia are the immune cells of the brain; the HIV neuropathogenesis is attributed to mitochondrial dysfunction, where the mitochondria interact with HIV viral proteins, resulting in a significant increase in pro-inflammatory cytokines that influence viral survival and replication. These data will help identify potential target(s) for the regulation of microglial activation and decrease of oxidative stress, thereby limiting microglia-associated neurotoxicity.

## 2. Materials and Methods

Study Design:

In vitro: Microglial cells were treated with HIV Tat (100 ng) or cART (300 µM TFV + 50 µM FTC + 188 nM DTG) for 3–24 h, followed by gene expression analysis, quantification of mitochondrial respiration, ROS and Raman spectroscopic analysis.

In vivo: Banked PBMC samples (*n* = 10/group) from HIV patients who had uncontrolled HIV with detectable viral load on nucleoside analogue containing regimen (cART = TFV, FTC, DTG) and HIV patients who had uncontrolled HIV with detectable viral load but were treatment naïve (control group) were used in the study. Informed consent was obtained from all subjects at enrollment for use of banked PBMC, and the study was conducted only after approval was obtained from the institutional IRB. Samples were de-identified and therefore no identifiable patient information was available. Subjects in the two study groups were age and sex matched.

### 2.1. Cell Culture

The HTHU (transformed human µglia) and the HTHU/HIV microglial cell lines were a generous donation from Dr. Jonathan Karn (CWRU, Cleveland, OH, USA). The transformed immortalized human microglial cells expressed key microglial surface markers. The HTHU cell line was generated from cryopreserved human microglia (Sciencell Cat # 1900) and infected with vesicular stomatitis Virus G envelope Simian Virus 40 large T antigen viral particles (VSVG SV40) containing the pBABE-puro SV40 CT construct (Addgene Plasmid # 13970) and by spinoculation. These transformed human microglia were allowed to expand in the presence of the selection antibiotic, puromycin (2 ug/mL) [15]. HTHU/HIV immortalized human microglial cells bear an HIV construct as a proxy for HIV infected primary microglia. Both HTHU and the HTHU/HIV microglial cell lines were cultured in DMEM/F12 medium (Sigma-Aldrich, Cat No. D8437) supplemented with 5% fetal bovine serum (FBS), 100 U/mL penicillin and 100 μg/mL streptomycin. Cultures were maintained at 37 °C in humidified 5% CO_2_ in an incubator.

### 2.2. Cell Viability Assay

Cell viability was measured using the CCK-8 Kit (Dojindo Molecular Technologies, Inc., Rockville, MD, USA). The assay uses a water-soluble tetrazolium salt, WST-8, which is reduced by dehydrogenase activity in cells to produce a yellow-color formazan dye soluble in the tissue culture media. The amount of the formazan dye generated by the dehydrogenase activity in cells is directly proportional to the number of living cells. HTHU and the HTHU/HIV cells (100,000 cells) were treated with Tat (10–100 ng/mL) for 48 h, and cell viability was measured using the CCK-8 Kit.3.

### 2.3. RNA Extraction

Cytoplasmic RNA was extracted by an acid guanidinium thiocyanate–phenol–chloroform method using Trizol reagent (Invitrogen-Life Technologies, Carlsbad, CA, USA). The amount of RNA was quantified using a Nano-Drop ND-1000 spectrophotometer (Nano-Drop™, Wilmington, DE, USA), and isolated RNA was stored at −80 °C until used.

### 2.4. Gene Expression Analysis Using Real Time PCR (QPCR)

Five hundred nanograms of total RNA was used for the RT reaction (25 μL total volume) with the First-Strand cDNA Synthesis Kit (GE Healthcare, Piscataway, NJ, USA), according to the manufacturer’s instructions. One microliter of the resultant cDNA from the RT reaction was employed as the template in PCR reactions using well validated primers obtained from Integrated DNA technologies (IDT). The primer sequences used were TRF1 (forward) 5′-CCACATGATGGAGAAAATTAAGAGTTAT-3′, (reverse) 5′-TGCCGCTGCCTTCATTAGA-3′; IL-1β (forward) 5′-CTCTCACCTCTCCTACTCACTT-3′ (reverse), 5′-TCAGAATGTGGGAGCGAATG-3′; Telomerase (forward) 5′-CGTCGAGCTGCTCAGGTCTT, (reverse) 5′-AGTGCTGTCTGATTCCAATGCTT, -3′. QPCR was performed using the Stratagene 3005P qPCR machine (Agilent, Santa Clara, CA, USA). A standard PCR reaction was carried out using the Power^®^ SYBR Green PCR Master Mix (Applied Biosystems, Foster City, CA, USA) and validated primers. The final primer concentration used in the PCR was 0.1 µM. PCR conditions were as follows: 95 °C for 3 min, followed by 24 cycles of 95 °C for 40 s, 58 °C for 30 s and 72 °C for 1 min; the final extension was at 72 °C for 5 min. Ct values were obtained for the gene of interest and the housekeeping gene beta actin. Gene expression levels were quantified using the comparative CT method [16]. The threshold cycle (Ct) of each sample was determined, the relative level of a transcript (2ΔCt) was calculated by obtaining ΔCt (test Ct−GAPDH Ct), and transcript accumulation index (TAI) was calculated as TAI = 2^−ΔΔCT^ [16].

### 2.5. Analysis for Telomere Length

We used the relative human telomere length quantification qPCR assay kit from ScienCell Inc. (RHTLQ cat # 8908, ScienCell Inc., Carlsbad, CA, USA) for TL analysis. QPCR data was analyzed using the Comparative CT method, where relative telomere length was calculated based on the formula T/S = 2^−ΔΔCt^, where ΔCt = Ct telomere-Ct SCR or (single copy reference) primer. The relative telomere length was calculated for the samples from patients on cART vs. patients who were cART naïve, and also the cDNA isolated from HTHU vs. the HTHU/HIV cells.

### 2.6. Quantification of Mitochondrial DNA (mtDNA)

We isolated DNA from PBMC of patients on cART and patients who were cART naïve using the Quick-DNA™ Universal Kit (Cat # D4068, Zymo Research, Irvine, CA, USA) and quantified mtDNA using real time PCR with the following human primers: for the target gene tRNA-Leu (UUR) (forward primer 5′-CAC CCA AGA ACA GGG TTT GT-3′ and reverse primer 5′-TGG CCA TGG GTA TGT TGT TA-3′) and nuclear β2-microglobulin (forward primer 5′-TGC TGT CTC CAT GTT TGA TGT ATC T-3′ and reverse primer 5′-TCT CTG CTC CCC ACC TCT AAG T-3′), respectively. The PCR reaction was as follows: 2 μL of template DNA (3 ng/μL isolated DNA), 2 μL of mtDNA target specific primer pair (400 nM final concentrations each), 12.5 μL SYBR Green PCR Master Mix (Roche Diagnostics Operations, Inc, Indianapolis IN) and 8.5 μL H_2_O/well, respectively. PCR was performed using the MX3005P (Stratagene, La Jolla, CA, USA). Samples were run in triplicate. We obtained both mtDNA and nucDNA CT average values, and determined the mtDNA content relative to nuclear DNA.

### 2.7. Mitochondrial Respiration

Microglial mitochondrial energetics was determined using the XFe24 extracellular flux analyzer (Seahorse Bioscience, North Billerica, MA, USA). The HTHU microglial cells were seeded in a Seahorse 24-well tissue culture plate at an optimized concentration of 50,000 cells per well. To evaluate the impact of HIV Tat (100 ng) and cART (300 µM TFV + 50 µM FTC + 188 nM DTG) on microglial mitochondrial respiration, these drugs were added to the cells 24 h prior to measurement. Basal oxygen consumption rate (OCR) and extracellular acidification rate (ECAR) were detected in the presence of physiological concentrations of 10 mM glucose, 1 mM pyruvate and 2 mM glutamine. Next, mitochondrial ATP dependent respiration, uncoupled respiration, and non-mitochondrial respiration were assessed after the addition of 1 µM oligomycin, 1.5 µM carbonylcyanide-p-trifluoromethoxyphenylhydrazone (FCCP) and the combination of 0.5 µM each of antimycin A and rotenone, respectively. All experiments were performed using 5 wells per treatment.

### 2.8. Immunofluorescent Staining

Microglia were grown to 70% confluence in a glass petri dish and treated with HIV- Tat (100 ng) for 24 h. Standard immunofluorescent staining procedures were followed. Cells were fixed for 10 min at RT using 4% formaldehyde, followed by permeabilization with ice-cold 90% methanol. Cells were then washed in 1× phosphate buffered saline (PBS) and treated with primary antibodies against TRF-1 (Anti-TRF1 Antibody (G-7), cat # sc-271485) and IL1β (Anti-IL-1β Antibody (11E5) cat # sc-52012), obtained from Santa Cruz Biotechnology, Inc. (Dallas, TX, USA). The secondary antibodies used include fluorescence labeled Alexa Fluor^®^ 488 Anti-mouse Secondary Antibodies, obtained from Thermo Fisher Scientific, Grand Island, NY, USA. The expression levels of TRF-1 and IL-1β were quantified with respect to an untreated control, based on the intensity of the fluorescent signal analyzed using the computer image analysis ImageJ software (ImajeJ 1.38e, https://imagej.nih.gov/ij/ (accessed on 25 June 2021), 1997–2018, National Institutes of Health, Bethesda, MA, USA). Imaging was performed with the EVOS^®^ FL Cell Imaging System (Life Technologies, Grand Island, NY, USA).

### 2.9. Reactive Oxygen Species (ROS) Measurement

ROS was quantified using the general oxidative stress indicator reagent from Invitrogen™ CM-H2DCFDA (cat # C6827). CM-H2DCFDA is a chloromethyl derivative of H2DCFDA, useful as an indicator for reactive oxygen species in cells. CM-H2DCFDA passively diffuses into cells, where its acetate groups are cleaved by intracellular esterases and its thiol-reactive chloromethyl group reacts with intracellular glutathione and other thiols, and subsequent oxidation yields a fluorescent product that can be quantified using fluorescent microscopy imaging. HTHU cells were plated on glass-bottom petri dishes and cells grew to 80% confluence. Cells were then treated with cART (300 µM TFV + 50 µM FTC + 188 nM DTG) for 24 h, following which the cells were washed with 1× PBS, following which 5 μM of CM-H2DCFDA (freshly prepared in HBSS) was added, and cells were incubated for 30 min in a dark CO_2_ incubator. After 30 min, the cells were washed with PBS, and the ROS production was quantified immediately by measuring the GFP fluorescence using EVOS^®^ FL Cell Imaging System (Life Technologies, Grand Island, NY, USA). Negative controls were assessed as follows: unstained cells were examined for autofluorescence in the green emission range. Measurements were obtained using excitation sources and filters appropriate for fluorescein Ex/Em: ~492–495/517–527 nm.

### 2.10. Raman Spectroscopy

Raman spectra were acquired with the HORIBA XploRA PLUS Raman microspectroscope. Excitation wavelength of 532 nm was used. Cells were kept in the incubator and removed for measurements at baseline and 24 h post treatment with cART (300 µM TFV + 50 µM FTC + 188 nM DTG). HTHU and HTHU/HIV cells were grown on quartz substrates to minimize fluorescence. For each time point, 3 measurements of 15 accumulations of 30 s were collected. Entrance slit of 50 µm yielded an approximate spectral resolution of 4 cm^−1^. Raman signal was collected by spectrometer equipped with a 1024 × 256 TE air cooled CCD chip (pixel size 26 micron, temperature −60 °C). HORIBA LabSpec6 software (version 6.4.2) was used for data acquisition, fluorescent background removal, baseline correction, and peak fitting.

### 2.11. Statistical Analysis

The data were analyzed using Graph Pad Prism 5.0 (GraphPad Software, La Jolla, CA, USA). Results were expressed as mean ± SD; for comparisons between 2 groups, the “*t*-test” was used for parametric data, and Mann–Whitney test for nonparametric data. A *p* value of <0.05 was considered as a statistically significant difference.

## 3. Results

### 3.1. Effect of cART on TL and Telomerase/TRF-1 Gene Expression in HIV Patients

We examined the TL and Telomerase/TRF-1 gene expression in uncontrolled HIV patients on cART and those who were cART naïve (a total of *n* = 10 patients/group). RNA was previously obtained and stored at −80 °C before being used to quantify TL and telomerase/TRF-1 gene expression by QPCR. We observed a 40% decrease (*p* < 0.05) in telomere length, a 47% increase (*p* < 0.05) in telomerase expression and a 37% increase (*p* < 0.05) in TRF-1 gene expression in RP on cART as compared to RP’s who were cART naïve (Figure 1A–D).

### 3.2. Effect of cART on mtDNA Expression in HIV Patients

The metabolic change in a cell is a result of a shift in homeostatic balance in cellular respiration and is dictated by expression of the mitochondrial genome (mtDNA). We examined the mtDNA expression in DNA extracted from PBMC samples obtained from a total of *n* = 10 patients/group HIV-1 uncontrolled patients on cART and those who were cART naïve. Our results showed a 121% increase (*p* < 0.01) in mtDNA expression in HIV-infected uncontrolled patients on cART as compared to RP’s who were cART naïve (Figure 1D).

### 3.3. Effect of HIV Tat and cART on TL in Microglia

Both HTHU and HTHU/HIV cells (100,000 cells/mL) were treated with HIV Tat (10 and 100 ng/mL) and cART (300 µM TFV + 50 µM FTC + 188 nM DTG) for 24 h, followed by RNA extraction, reverse transcription and QPCR. Our results (Figure 2) showed that HTHU treated with 10 ng/mL and 100 ng/mL of HIV-Tat resulted in an 18% (*p* = NS) and 25% (*p* = NS) decrease in telomere length, respectively. On the other hand, HTHU/HIV microglial cells treated with 10 ng/mL and 100 ng/mL of HIV-Tat resulted in an 18% (*p* = NS) and 58% (*p* < 0.05) decrease in telomere length, respectively. HTHU cells treated with cART resulted in a 43% decrease (*p* < 0.05) and HTHU/HIV cells treated with cART resulted in a 77% decrease (*p* < 0.01) in telomere length, respectively.

### 3.4. Effect of HIV Tat and cART on Telomerase Gene Expression in Microglia

Both HTHU and HTHU/HIV cells (100,000 cells/mL) were treated with HIV Tat (10 and 100 ng/mL) and cART (300 µM TFV + 50 µM FTC + 188 nM DTG) for 24 h, followed by RNA extraction, reverse transcription and QPCR. Our results (Figure 3) showed that HTHU treated with 10 ng/mL and 100 ng/mL of HIV-Tat did not produce any significant change in telomere gene expression, as compared to the untreated control. HTHU cells treated with cART also did not show a significant decrease in telomerase gene expression. On the other hand, HTHU/HIV microglial cells treated with 10 ng/m and 100 ng/mL of HIV-Tat resulted in a 59% (*p* < 0.05) and 62% (*p* < 0.05) decrease in telomerase gene expression, respectively, while treatment with cART in these cells resulted in a 74% decrease (*p* < 0.01) in telomerase gene expression as compared to the untreated control.

### 3.5. Effect of HIV Tat on TRF-1 Expression in Microglia

We examined the expression of TRF1 in human primary microglial cells (HTHU) treated with 100 ng/mL HIV Tat. Using immunofluorescence staining, we observed a 2.1-fold increase (*p* < 0.01) in the expression of telomere repeat-binding factor 1 (TRF1) in HTHU cells treated with 100 ng/mL HIV Tat (Figure 4A). We also quantitated TRF-1 gene expression using QPCR and observed a 37% increase (TAI = 1.37 ± 0.036; *p* < 0.05) in TRF-1 gene expression in HIV Tat (100 ng/mL) treated HTHU cells as compared to the untreated control (TAI = 1.0 ± 0.08) (Figure 4C). These data suggest a correlation between TRF1 overexpression and telomere shortening in HIV Tat treated microglia.

### 3.6. Effect of HIV Tat on Pro-Inflammatory Cytokine IL-1β Expression in Microglia

We examined the expression of IL-1β in human primary microglial cells (HTHU) treated with 100 ng/mL HIV Tat. Using immunofluorescence staining, we examined the expression of IL-1β in human primary microglial cells (HTHU) treated with 100 ng/mL HIV Tat. We observed a 2.32-fold increase (*p* < 0.01) in the expression of IL-1β in HTHU cells treated with 100 ng/mL HIV Tat (Figure 4B). We also quantitated IL-1β gene expression using QPCR and observed a 74% increase (TAI = 1.74 ± 0.11; *p* < 0.01) in TRF-1 gene expression in HIV Tat (100 ng/mL) treated HTHU cells as compared to the untreated control (TAI = 1.0 ± 0.06) (Figure 4C). These data suggest that HIV Tat treatment induced a significant pro-inflammatory response in HIV Tat treated microglia.

### 3.7. Effect of HIV Tat and cART on Mitochondrial Energetics

We examined the effect of HIV Tat/cART on mitochondrial stress. Microglial mitochondrial energetics was determined using the XFe24 extracellular flux analyzer (Seahorse Bioscience, North Billerica, MA, USA). The microglial cells (HTHU) were seeded in a Seahorse 24-well tissue culture plate at an optimized concentration of 30,000 cells per well. To evaluate the impact of HIV Tat on microglial mitochondrial respiration, 100 ng/mL Tat was added to the cells 24 h prior to measurement. To evaluate the impact of cART on microglial mitochondrial respiration, cART (300 µM TFV + 50 µM FTC + 188 nM DTG) was added to the cells 24 h prior to measurement. Basal oxygen consumption rate (OCR) and extracellular acidification rate (ECAR) were detected in the presence of physiological concentrations of 10 mM glucose, 1 mM pyruvate and 2 mM glutamine. Next, mitochondrial ATP dependent respiration, uncoupled respiration and non-mitochondrial respiration were assessed after the addition of 1 µM oligomycin, 1.5 µM carbonylcyanide-p-trifluoromethoxyphenylhydrazone (FCCP) and the combination of 0.5 µM each of antimycin A and rotenone, respectively. All experiments were performed using five wells per treatment. Figure 5A shows the cellular energetics profile, which provides an overview of mitochondrial function.

As shown in Figure 5B, both 100 ng/mL Tat and cART treatment resulted in a significant reduction in mitochondrial respiration. Figure 5C shows that basal OCR was decreased by 69% and 74% in HTHU microglial cells in response to 100 ng/mL Tat and cART treatment, respectively. This occured due to a decreased spare respiratory capacity (Figure 5C) in both proton leak OCR and ATP-linked OCR (Figure 5D). Mitochondrial dysfunction occurs when oxidative modification of the respiratory chain complexes occurs, and this event amplifies and promotes further oxidative damage. Our results show that HIV Tat and cART impair mitochondrial respiration in microglia, as measured by maximal and spared oxygen consumption, which could lead to bioenergetic dysfunction and production of ROS.

### 3.8. Effect of HIV Tat on ROS Production

Figure 6A–D shows that a significant increase in ROS production was observed in both the HTHU cells treated with Tat (1.7-fold increase, *p* < 0.05) and cART (1.5-fold increase, *p* < 0.05), as compared to the untreated control. The fluorescent intensity was quantified by ImageJ (https://imagej.nih.gov/ij/ (accessed on 27 June 2021)) and expressed in mean pixel units. It reflects increased oxidative stress that eventually contributes to neuro-inflammation and potential neurocognitive impairment.

### 3.9. Effect of cART on Metabolic Changes in HTHU and HTHU/HIV Microglia Quantified by Raman Spectroscopy

Raman spectroscopy is often used on complex biological samples to identify chemical signatures and changes produced by alterations in the environment, such as through cell treatment [17,18,19,20,21,22,23,24,25,26]. Inelastic Raman scattering, which occurs when there is an energy exchange between an incident photon and a molecule, results in scattered photons of different wavelengths than the original ones. These photons contain information revealing the chemical structure of the sample. The main advantage of Raman spectroscopy is that it does not require staining, fluorescent markers or any other sample processing that can unintentionally modify the sample. Another advantage of this technique is that it is not a specialized test, as it detects all the changes in the samples as a result of a treatment.

Raman spectroscopy was used to profile metabolic changes in HTHU and HTHU/HIV microglia treated with cART to identify any alterations in proteins, phospholipids and nucleic acids in the cART treated microglia. For both HTHU and HTHU/HIV cells, the peak intensity after treatment with cART was lower than at baseline, respectively (Figure 7A,B). Our data identified reductions of the peak intensities in the spectral regions associated with glucose (1124 cm^−1^), lipids/phospholipids (1116 cm^−1^, 1098 cm^−1^, 1077 cm^−1^), proteins (1120 cm^−1^), nucleic acids (1081 cm^−1^) and phenylalanine (1103 cm^−1^) upon cART treatment. These changes indicate a broad range of cell dysfunction. The results obtained in this study provide insights into understanding the metabolic effects of HIV therapy.

## 4. Discussion

HIV can enter the CNS during early stages of infection, and persistent CNS HIV infection and inflammation probably contribute to the development of HIV-associated neurocognitive disorders (HAND). The brain can subsequently serve as a sanctuary for ongoing HIV replication, even when systemic viral suppression has been achieved [27]. Fifty percent of patients treated with cART have milder forms of HAND, such as asymptomatic neurocognitive impairment (ANI) and mild neurocognitive disorder (MND) [28]. HAND generally remains stable during cART, but rarely resolves completely, therefore the real impact of cART on HAND remains ill-defined [29,30].

Telomeres and telomerase activity have been characterized as biomarkers of cellular aging [13]. Telomeres are coated with a protein complex consisting of a group of proteins, such as telomere repeat-binding factor 1 (TRF1) and 2 (TRF2), which prevent the DNA repair machinery from recognizing and processing telomeres during the repair of double-stranded DNA breaks. When telomere length becomes critically short, proliferation is arrested, and the risk of apoptosis is increased. Critically short telomeres can cause telomere dysfunction, which eventually induces DNA damage responses at the telomeres [31]. Proliferating cells produce telomerase, an enzyme that acts as a reverse-transcriptase, and is responsible for catalyzing the addition of nucleotides using an RNA template [32]. Telomerase is a cellular reverse transcriptase responsible for the de novo synthesis of telomeric DNA; it requires reverse transcriptase to elongate the telomere [33]. In HIV-infected patients, telomere shortening is caused by inhibition of human telomerase by antiretroviral drugs, more specifically NRTIs [13,33,34,35]. Tenofovir (TFV) at therapeutic concentrations is a potent inhibitor of telomerase activity, causing telomere shortening in vitro [13,34,35,36]. TFV is a more potent inhibitor of telomerase than abacavir, lamivudine or emtricitabine [13,36]. NRTIs, including zidovudine, stavudine, tenofovir, didanosine and abacavir, inhibit telomerase effectively in vitro [12]. Telomerase activity is also inhibited by currently recommended RTI agents, such as tenofovir and abacavir [12,13].

Despite the recognition of a role of mitochondrial toxicity in HIV and cART complications for several decades, the pathophysiology of these complications is still poorly understood. Oxidative damage generated during oxidative phosphorylation of mitochondrial macromolecules, such as mtDNA, and alteration in mitochondrial biogenesis, may be responsible for accelerated neuropathology in HIV patients. To our knowledge, limited studies have explored the effects of combination antiretroviral therapy (cART), such as nucleoside-based regimen TAF, FTC and DTG, on TL and the underlying mechanisms such as oxidative stress and mitochondrial dysfunction which may drive these effects in HIV-associated neurocognitive impairments in HIV-infected patients receiving these cART regimens. Our results (Figure 1A–C) show a 40% decrease (*p* < 0.05) in telomere length, a 47% increase (*p* < 0.05) in telomerase expression and a 37% increase (*p* < 0.05) in TRF-1 gene expression in infected HIV uncontrolled patients on cART, compared to infected HIV uncontrolled patients who were cART naïve, indicating that telomere homeostasis is regulated by telomerase and associated proteins, such as TRF1. TRF1 acts as a negative regulator of telomere length by inhibiting telomerase activity; overexpression of TRF1 in telomerase-positive cells results in a gradual telomere shortening, similar to observations reported by Muñoz et al. in 2009 [37].

Neuroinflammatory response is regulated by microglia, the resident immune cells of the CNS. Under normal circumstances the resting microglia play a pivotal role in maintaining tissue homeostasis and promoting brain development, however, upon activation by HIV, microglia secrete high levels of proinflammatory factors and the overaccumulation of these factors causes neuronal apoptosis and subsequent cognitive impairment, such as in HAND. Microglia are the only resident cells in the brain parenchyma that can support productive HIV infection, and therefore are likely to be a major contributor to neurotoxicity observed during HIV infection. Our in-vitro microglial culture experiments showed significant decreases in TL in both HIV transfected and untransfected microglia (Figure 2). We observed a significant decrease in the telomerase gene (Figure 3) only in HIV transfected microglial cells treated with HIV viral protein Tat and cART compared to the untreated control, but no significant differences were observed in microglial cells that were untransfected. Our data and reports from other studies suggest that HIV Tat and cART induced changes in TL and telomerase activity that may directly contribute to microglial cellular senescence and underlie the neurodegenerative effects observed in HAND [38,39]. A recent study by Kronenberg, G. et al. suggested that classical microglial activation is associated with suppression of telomere-associated genes and may underlie microglia dysfunction [40].

Mitochondria play a critical role in the regulation of apoptosis. Therefore, mitochondrial oxidative stress contributes to neuronal apoptosis. Interpretation of studies assessing mitochondrial complications have been limited by lack of consensus on the optimal in vivo and ex vivo measure(s) by which to quantify mitochondrial dysfunction. Chronic HIV infection and inflammation and/or cART drugs have adverse effects on mitochondrial function, which would contribute to long-term complications in HIV-infected persons. Patients with HIV associated comorbidities may require additional pharmacologic interventions that can complicate therapeutic management. Older patients with HIV are at an even greater risk of polypharmacy-related adverse drug reactions from both ARV drugs and other concomitantly administered medications. We examined mtDNA in DNA from PBMC of infected HIV uncontrolled patients on cART and patients who were cART naïve. Our results showed a 121% increase (*p* < 0.01) in mtDNA expression in HIV infected and HIV uncontrolled patients on cART as compared to patients who were cART naïve (Figure 1D). Since mtDNA encodes vital components of the OXPHOS and protein synthesis machinery, oxidative damage-induced mtDNA mutations that impair either the assembly or the function of the respiratory chain will, in turn, cause further accumulation of ROS, which results in a vicious cycle leading to energy depletion in the cell, and ultimately cell death. Our results suggests that in addition to HIV viral protein, cART treatment further contributes to mitochondrial dysfunction.

Our mitochondrial biogenesis assessment by the Seahorse assay showed that HIV Tat and cART impairs mitochondrial respiration in microglia, as measured by maximal and spared oxygen consumption, which could lead to bioenergetic dysfunction and production of ROS. The role of oxidative stress in the pathophysiology of accelerated aging in HIV is well established: reactive oxygen species (ROS) induced oxidation results in cell dysfunction and apoptosis. The production of ROS is common under oxidative stress induced aging, and we speculate that HIV subjects may develop accelerated aging because of an impaired mitochondrial function that induces an early senescence [41,42]. We showed a significant increase in ROS production in HTHU microglia treated with Tat and cART, suggesting that under non-activated conditions, microglia are likely to rely on oxidative metabolism. However, upon stimulation, microglia may switch from oxidative metabolism to glycolytic metabolism to support their shift in activation state. Mitochondria play a central role in the HIV neuropathogenesis process, since it is both a major ROS producer, and a target of ROS induced dysfunction. The proximity of mtDNA to the ROS-generating electron transport chain makes mtDNA susceptible to oxidative damage. The role of metabolic reprogramming in the regulation of innate immune responses is under investigation, and microglial metabolism and mitochondrial dysfunction, in the context of HIV infection, is largely understudied.

Evidence from other studies shows that activated microglial cells (BV-2) exhibit decreased oxygen consumption rates and increased lactate release, highlighting the shift from oxidative metabolism towards glycolytic metabolism [43,44,45]. It is proposed that during this shift, cells preferentially utilize glycolysis rather than oxidative phosphorylation in an effort to preserve and generate the metabolic resources needed to meet the demands associated with cellular proliferation and activation, while still producing a sufficient supply of ATP [45,46]. HIV Tat has been reported to significantly reduce mitochondrial membrane potential in mouse primary microglia, and these reductions in cellular energetics are evidenced by decreased basal and maximal respiration, decreased ATP production and reduced extracellular acidification, which is a measure of glycolysis [47]. These reports and our results confirm that Tat mediates mitochondrial dysfunction in microglia. Further, mitochondrial dysfunction triggers an inflammatory response in microglia [43,48,49], and thus Tat-mediated mitochondrial dysfunction may fuel chronic inflammation in response to HIV infection of microglia. Our Raman spectroscopy data suggest that both HIV Tat and cART significantly alter proteins, phospholipids and nucleic acids in microglia, which supports our hypothesis. Additional studies are needed to explore the impact these compounds have on brain metabolism and subsequently cognition.

## 5. Conclusions

Accelerated telomere shortening contributes to genetic instability of cells, cellular senescence, cell cycle arrest or apoptosis that may contribute to the neurodegeneration and cognitive deficits typically observed in HAND patients. Telomere shortening occurs early following HIV acquisition and there is a persistent decline of telomere length indicating that telomeres are involved in the pathogenesis and clinical progression of HIV, however, the long-term biological significance of such changes are unknown.

Overall, our data suggest that both HIV Tat and cART induce metabolic stress in the microglia, resulting in persistent microglial activation and leading to mitochondrial dysfunction, which could be the underlying mechanism in the neurocognitive impairment observed in HAND patients, both in the untreated HIV patients as well as in the cART suppressed HIV patients. It is well documented that cART regimens containing NRTIs are potent inhibitors of telomerase activity, which we also observed in our study. Our future investigations will examine the effect of a switch in ART regimen from NRTI to NNRTI and its impact on telomere length in microglia and the associated biological relevance to neuro-pathogenesis of HIV.

## Figures and Tables

**Figure 1 vaccines-09-00721-f001:**
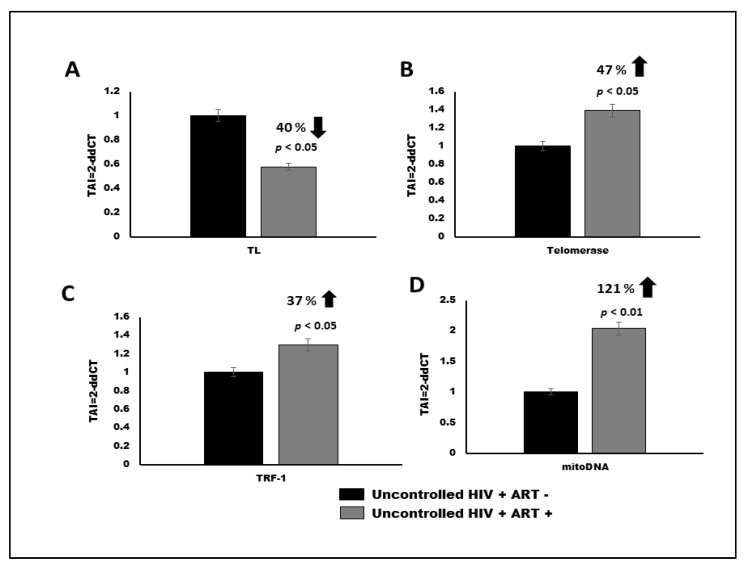
(**A**–**D**): Quantitation of telomere length, telomerase, TRF-1 gene expression and mtDNA by QPCR between HIV positive patients with uncontrolled infection who were on cART vs. treatment naïve patients. Results are expressed as the mean ± SD from a total of (*n* = 10 subjects/group). A *p* value of <0.05 is considered a statistically significant difference.

**Figure 2 vaccines-09-00721-f002:**
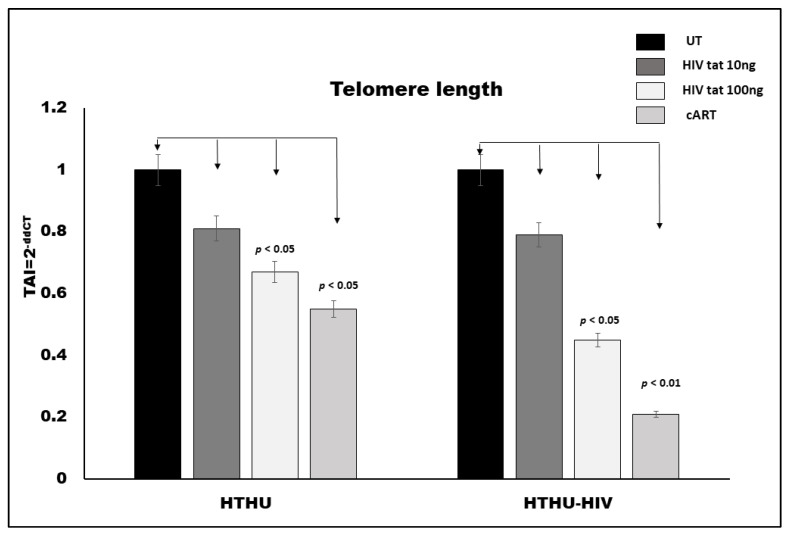
Effect of HIV-tat and cART on telomere length expression in HTHU and HTHU/HIV as quantitated by QPCR. Results are expressed as the mean ± SD from separate experiments (*n* = 3) done in triplicate. A *p* value of <0.05 is considered a statistically significant difference.

**Figure 3 vaccines-09-00721-f003:**
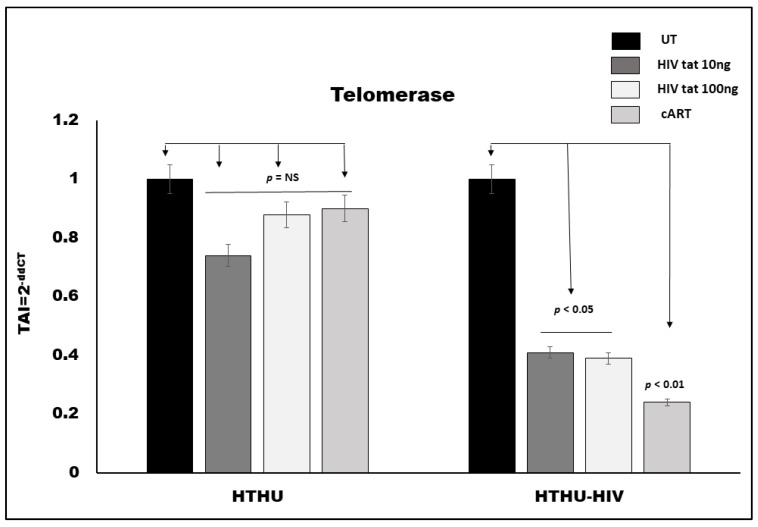
Effect of HIV-tat and cART on telomerase gene expression in HTHU and HTHU/HIV as quantitated by QPCR. Results are expressed as the mean ± SD from separate experiments (*n* = 3) done in triplicate. A *p* value of <0.05 is considered a statistically significant difference.

**Figure 4 vaccines-09-00721-f004:**
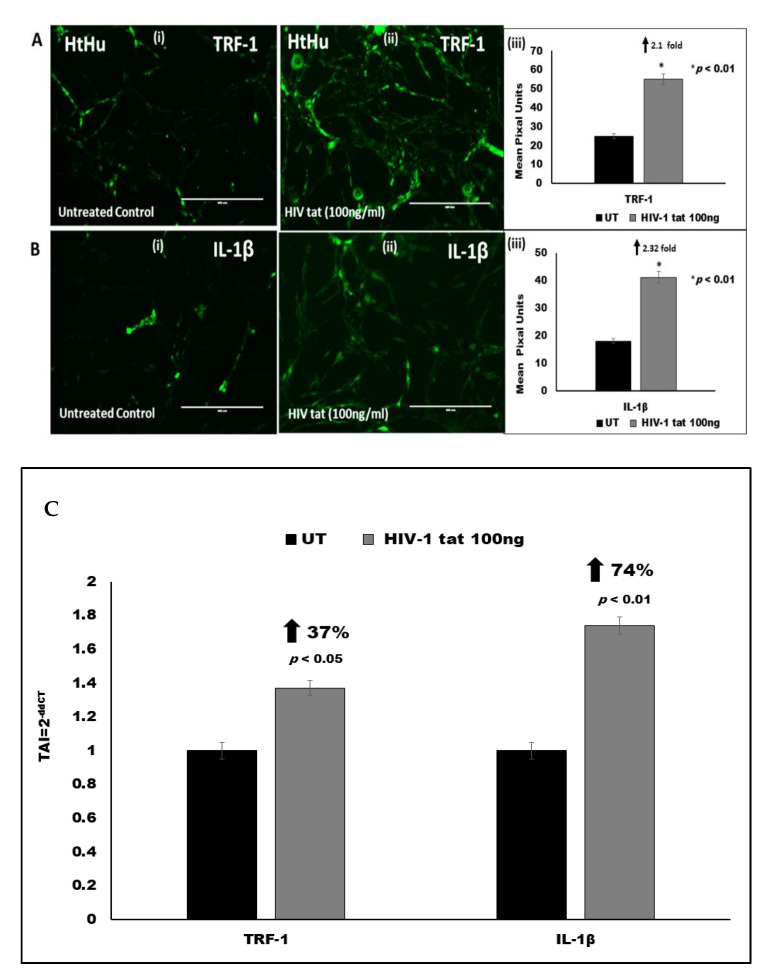
Effect of HIV-tat on TRF-1 and IL1β expression in HTHU microglial cells. (**A**): Immunostaining for TRF-1 expression (green) stained with Alexa Fluor 488 dye; (**A**(**i**)): Untreated control; (**A**(**ii**)): HIV tat (100 ng/mL) treated microglial cells; (**A**(**iii**)): Representative histogram of quantitation of fluorescent signal showing TRF-1 expression by Image J. (**B**): Immunostaining for IL-1β expression (green) stained with Alexa Fluor 488 dye; (**B**(**i**)): Untreated control; (**B**(**ii**)): HIV tat (100 ng/mL) treated microglial cells; (**B**(**iii**)): Representative histogram of quantitation of fluorescent signal showing IL-1β expression by Image J. Expression was quantitated using ImageJ software and statistical analysis was performed based on comparison between treated and untreated control. A *p* value of <0.05 is considered a statistically significant difference. Standard immunostaining protocols were followed. Results shown are representative images from (*n* = 3) separate experiments. (**C**): Effect of HIV-tat and cART on Telomerase gene expression in HTHU and HTHU/HIV as quantitated by QPCR. Results are expressed as the mean ± SD from separate experiments (*n* = 3) done in triplicate. A *p* value of <0.05 is considered a statistically significant difference.

**Figure 5 vaccines-09-00721-f005:**
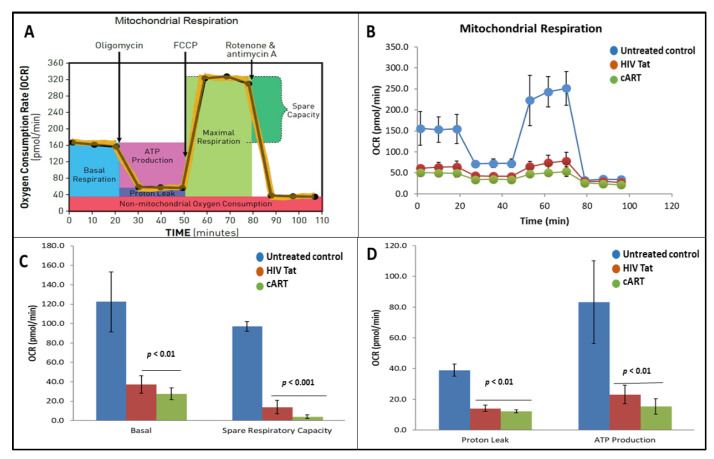
Effect of HIV tat and cART on Cellular Energetics as assessed using the Seahorse assay. **Panel** (**A**) shows a generalized cellular energetics profile, which provides an overview of mitochondrial function. The basal OCR is derived by subtracting non-mitochondrial respiration, which is subtracted across the entire experiment. Treatment with oligomycin, a complex V inhibitor, demonstrates the proportion of ATP-linked OCR in the basal OCR by subtracting the oligomycin rate from the basal OCR. The proton leak respiration is derived by subtracting non-mitochondrial respiration from the oligomycin rate. FCCP, an uncoupling agent, is added to allow the electron transport chain (ETC) to function at its maximal rate and the maximal respiratory capacity is derived by subtracting non-mitochondrial respiration from the FCCP rate. Antimycin A and rotenone are added to shut down ETC function, which allows calculating the non-mitochondrial respiration. The mitochondrial reserve capacity is calculated by subtracting basal respiration from maximal respiratory capacity. **Panel** (**B**) shows the cellular energetics profile at 24 h post treatment for microglia treated with HIV tat and cART. **Panel** (**C**) shows both HIV tat and cART exposure decreased oxygen consumption rates, which is indicative of mitochondrial dysfunction. HIV tat and cART treated microglia cells demonstrated significantly decreased uncoupled mitochondrial respiration following the addition of FCCP evident from decreased cellular energetics. **Panel** (**D**) shows a decrease in proton leak and ATP production indicating a change in non-mitochondrial respiration. Statistical comparisons were performed based on the untreated control and a *p* value of <0.05 is considered a statistically significant difference.

**Figure 6 vaccines-09-00721-f006:**
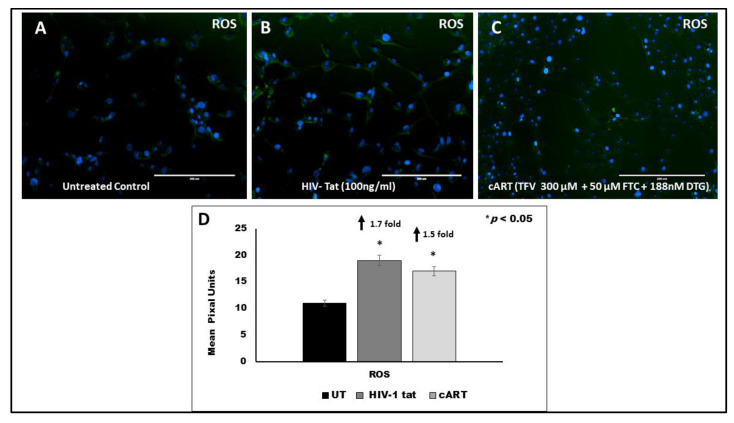
Effect of HIV tat and cART on ROS production in HTHU microglial cells was quantitated by measuring the green fluorescence product resulting from oxidation of CM-H2DCFDA. **Panel** (**A**): Untreated control; **Panel** (**B**): HIV tat (100 ng/mL) treated microglia cells; **Panel** (**C**): cART (300 µM TFV + 50 µM FTC + 188 nM DTG) treated microglia cells; **Panel** (**D**): Representative histogram of quantitation of fluorescent signal showing ROS expression by Image J. Expression was quantitated using Image J software. Results shown are representative images from (*n* = 3) separate experiments. GFP fluorescence was quantitated using EVOS^®^ FL Cell Imaging System (Life Technologies). Negative controls were assessed by examining unstained cells for autofluorescence in the green emission range. Measurements were obtained using excitation sources and filters appropriate for fluoresceine Ex/Em: ~492–495/517–527 nm. Statistical comparisons were made between treated and untreated control and a *p* value of <0.05 is considered a statistically significant difference. DAPI (blue) was used as a nuclear stain.

**Figure 7 vaccines-09-00721-f007:**
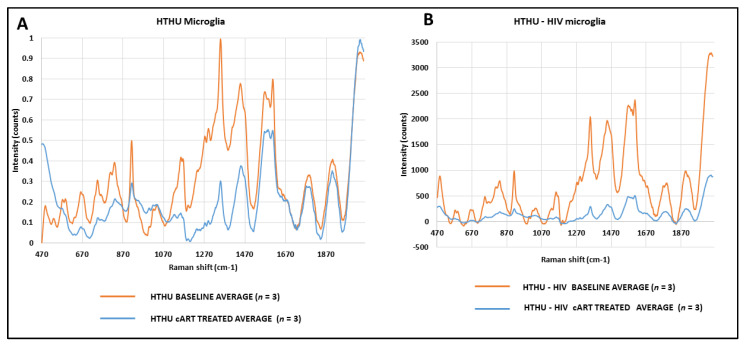
Raman spectra of HTHU and HTHU/HIV microglia treated with cART. A minimum of 3–4 Raman hyperspectral datasets will be obtained per sample, following which, the fluorescence background signal will be subtracted from the Raman spectra at each pixel in the image by a modified polyfit fluorescence removal technique using the HORIBA software. (**A**) HTHU Microglia; (**B**) HTHU—HIV microglia.

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
