# Peer review of "Telomere Length Shortening in Microglia: Implication for Accelerated Senescence and Neurocognitive Deficits in HIV"

_vaccines, 2021, doi:10.3390/vaccines9070721_

Round 1

Reviewer 1 Report

In this manuscript, authors Hsiao et al describe their studies on the telomere length and other biochemical properties of HIV patient derived, as well as microglial cells in tissue culture. Using a combination of qPCR, measurement of energetics and Raman spectroscopic techniques, the authors show that PBMCs from HIV patients on cART therapy, as well as microglial cells treated with the HIV protein Tat, show significant reduction of telomere length, reduced expression of TRF and telomerase, and increased oxidative stress and reduced mitochondrial DNA. Based on these studies the authors conclude that the neurocognitive defects seen in HIV patients on cART regimen may be due to these increased oxidative stress and reduced mitochondrial function.

Overall the research has been well designed, and data are compelling enough to make a strong case for the conclusions.

A few minor points of concern:

  • Language needs editing in several places. For example, in abstract, in first line, change “success” to “widespread use”. Similarly, “HIV infected results in” should be “infection results in”.
  • Some abbreviations are not expanded anywhere in the entire manuscript.

For example, please expand PBMC, HTHU. Also, in the figures, please specify that the label “UT” refers to “untreated”.

  • Figure legends need to be more descriptive. Also, each panel needs a few lines of description.
  • Figure 2, change the label “telomere length expression” to just telomere length.
  • Also clarify what the Y-axis label “TAI= 2ddCT” means.
  • Figure 4, labels on the bar graph are unclear.
  • Figure 5, panel A needs some description. Labels in panels B, C and D are too small. Also, what is meant by Veh is not clear.
  • In materials and methods, section 2.4, change sentence structure from “will be carried” to “was carried out”, or “Ct values will be obtained” to “were obtained”
  • Use of Raman spectroscopy, and the results don’t seem to add anything much to the manuscript, more than what is already shown by the other assays. Please explain exactly what was achieved in terms of inferences from this experiment.

Author Response

REV 1:

Overall the research has been well designed, and data are compelling enough to make a strong case for the conclusions.

  • We thank the reviewer for their encouraging comment.

A few minor points of concern:

  • Language needs editing in several places. For example, in abstract, in first line, change “success” to “widespread use”. Similarly, “HIV infected results in” should be “infection results in”
  • Some abbreviations are not expanded anywhere in the entire manuscript. For example, please expand PBMC, HTHU. Also, in the figures, please specify that the label “UT” refers to “untreated”
    • We have made these specific changes as suggested and have also reviewed the manuscript for language and editing errors and have corrected those. Additionally, we have included abbreviations that were previously missed such as PBMC and HTHU.
    • PBMC -Peripheral blood mononuclear cells
  • HTHU – (Transformed human  µglia) are  transformed immortalized human microglial  cells that  express key microglial surface markers.
  • Origin:Transformed, (SV40 large T antigen and hTERT)-

 Reference:  

  • Garcia-Mesa Y., Jay T. R., Checkley M. A., Luttge B., Dobrowolski C., Valadkhan S., et al. (2017). Immortalization of primary microglia: a new platform to study HIV regulation in the central nervous system. J. Neurovirol. 23 47–66.
  • Figure legends need to be more descriptive. Also, each panel needs a few lines of description.
    • All legends are now include additional description, that include information specific to each panel in the figure.
  • Figure 2, change the label “telomere length expression” to just telomere length.
    • This has been corrected
  • Also clarify what the Y-axis label “TAI= 2ddCT” means.
    • This is now explained in the legend and 
    • TAI is abbreviation for Transcript accumulation Index. TAI=2-ddCT is computed from PCR data as described in Ref# 16 by the comparative CT method.  This  method is a convenient way to analyze the relative changes in gene expression from real-time quantitative PCR experiments.  
    • Relative quantification relates the PCR signal of the target transcript in a treatment group to that of another sample such as an untreated control.
  • Figure 4, labels on the bar graph are unclear.
    • These are now re-sized and made larger, so labelling  is clear.
  • Figure 5, panel A needs some description. Labels in panels B, C and D are too small. Also, what is meant by Veh is not clear.
    • More description is now added to Panel A and all labels are now re-sized and made larger, so labelling is clear. Veh represented Vehicle or media control or untreated control. We have now removed the term Veh and have used the term untreated control for greater clarity.
  • In materials and methods, section 2.4, change sentence structure from “will be carried” to “was carried out”, or “Ct values will be obtained” to “were obtained”
    • This has been corrected
  • Use of Raman spectroscopy, and the results don’t seem to add anything much to the manuscript, more than what is already shown by the other assays. Please explain exactly what was achieved in terms of inferences from this experiment.
  • Overall, our data indicates that HIV tat and cART both induce metabolic stress and mitochondrial  dysfunction in microglia.  These in-vitro experiments are end point experiments, i.e. 24 hr post treatment. Raman experiments provided information about metabolic changes in real time. Raman spectroscopy enables molecular chemical analysis of living cells by comparing them to known Raman signatures of specific vibrational bonds. Our Raman data confirms that both  HIV tat and cART induces changes in protein & lipid composition, stress response, nuclear division  thus providing a  prognostic evaluation of mitochondrial function in HIV tat and cART treated microglia.

Reviewer 2 Report

                This study confirms and extends work on the effect of NRTIs and HIV infection on telomere length.  Overall the study is interesting, although it might be more appropriate for another journal as it does not discuss ‘Vaccines’ per se.  I do have several suggestions to clarify and polish the study/manuscript:

Major Points:

  1. 2:  Three points:  I’m impressed that telomere length can change that much in 24hrs (rather than requiring multiple rounds of cell division).  Please confirm that the experimental design is correct as stated.  Also what was the effect of tat expression on overall cell viability.  Finally, the shortest telomere length, rather than the average telomere length, is the best predictor for cellular senescence.  Optimally, this should be determined for the samples.
  2. 3: What was is the relative baseline expression of telomerase in HTHU/HIV verses uninfected cells?  It seems surprising that adding Tat to a cell that already has Tat from an infection would have more of an effect than adding Tat to an uninfected cell.  Also what was the effect of tat expression and NRTI treatment on overall cell viability in these experiments?  Just want to be certain that the data are reflective of direct rather than indirect effects.
  3. 4: was there a concomitant increase in the mRNA for TRF1 and IF1B?  For consistency with the rest of the data in the study, assaying mRNA via qRT-PCR would be informative.

Minor Points:

  1. Line 93:  Telomere should be lower case (telomere).  It is also randomly capitalized in other places throughout the manuscript – as are other words as well.  Please check the manuscript carefully in revision.
  2. Line 161: The Livak reference should be numbered as per journal style
  3. Lines 245-246: change: A p value  was < 0.05 was  to:  A p value of < 0.05 was
  4. Lines 253-254: fix the wording in this run on sentence for clarity:  which were previously obtained and were stored at – 800C were used to quantitate TL and…
  5. Line 262: statistically is spelled incorrectly here and in other figure legends
  6. 1: I’m intrigued by the tight error bars using these highly variable biological samples.  Just double checking that the standard deviations were calculated using biological and not technical replicates.
  7. Section 3.2: measuring mtDNA levels by QPCR is not really a measure of ‘mitochondrial DNA expression” (aka transcription) per se.  I would recommend rewording for clarity
  8. Line 282: telomere length is correct – not ‘telomere length expression’
  9. As implied by the above points, a carefully editing of the manuscript for spelling, capitalization and grammatical issues would be recommended.

Author Response

REV 2:

 This study confirms and extends work on the effect of NRTIs and HIV infection on telomere length.  Overall the study is interesting, although it might be more appropriate for another journal as it does not discuss ‘Vaccines’ per se.  I do have several suggestions to clarify and polish the study/manuscript:

  • We thank the reviewer for providing us with constructive comments that helped us improve the manuscript and provided us an opportunity to provide clarifications for some of the findings.

Major Points:

Three points:  I’m impressed that telomere length can change that much in 24hrs (rather than requiring multiple rounds of cell division).  Please confirm that the experimental design is correct as stated.  Also what was the effect of tat expression on overall cell viability.  Finally, the shortest telomere length, rather than the average telomere length, is the best predictor for cellular senescence.  Optimally, this should be determined for the samples

  • We believe that the rapid change in telomere length may perhaps be due to the fact that these activated microglia undergo a rapid cell division and dynamic transformation, and given that these are derived from primary microglia their telomeres shorten after every cell division.
  • Our experimental design for the Raman experiments is 3 hrs post treatment, while all other experimental paradigms are 24 hours post
  • The concentration of HIV tat used in our experiments was 100ng/ml.  No significant  change in cell viability with respect to the untreated control  is observed at this concentration. We  and other investigators have used a wide range ( 1ng-100ng/ml)  of HIV tat concentrations for  in-vitro  experiments and no cell toxicity is observed at 100ng/ ml Tat concentration. Cell toxicity is observed at HIV Tat concentration of 500ng/ml or greater. 

References:

  • Nath A, Conant K, Chen P, et al. Transient exposure to HIV-1 Tat protein results in cytokine production in macrophages and astrocytes. A hit and run phenomenon. J Biol Chem. 1999. June 11;274(24):17098–17102;
  • El-Hage N, Rodriguez M, Dever SM, et al. HIV-1 and morphine regulation of autophagy in microglia: limited interactions in the context of HIV-1 infection and opioid abuse. J Virol. 2015. January 15;89(2):1024–1035.
  • Chivero, E. T., Guo, M. L., Periyasamy, P., Liao, K., Callen, S. E., & Buch, S. (2017). HIV-1 Tat Primes and Activates Microglial NLRP3 Inflammasome-Mediated Neuroinflammation. The Journal of neuroscience : the official journal of the Society for Neuroscience, 37(13), 3599–3609.
  • Thangaraj, A., Periyasamy, P., Liao, K., Bendi, V. S., Callen, S., Pendyala, G., & Buch, S. (2018). HIV-1 TAT-mediated microglial activation: role of mitochondrial dysfunction and defective mitophagy. Autophagy, 14(9), 1596–1619.
  • Mahajan SD, Aalinkeel R, Sykes DE, Reynolds JL, Bindukumar B, Fernandez SF, Chawda R, Shanahan TC, Schwartz SA. Tight junction regulation by morphine and HIV-1 tat modulates blood-brain barrier permeability. J Clin Immunol. 2008 Sep;28(5):528-41.

  • We agree with the reviewer that the shortest telomere length, rather than the average telomere length, is the best predictor for cellular senescence and believe that with each cell division, telomeres shorten until they are unable to protect the chromosomes and  which then  leads to cell death. We are planning a future investigation that  will examine telomere shortening in an aging HIV cohort.

What was is the relative baseline expression of telomerase in HTHU/HIV verses uninfected cells?  It seems surprising that adding Tat to a cell that already has Tat from an infection would have more of an effect than adding Tat to an uninfected cell.  Also what was the effect of tat expression and NRTI treatment on overall cell viability in these experiments?  Just want to be certain that the data are reflective of direct rather than indirect effects.

  • The basal telomerase  expression in both the microglial cell lines was comparable as evident by the similar average  baseline CT value of   1 (HTHU)  and  32.4 ( HTHU/HIV) for untreated cells.   Both are immortalized   cell line derived from primary microglia cells and transformed using similar vectors except that the HTHU/HIV cell line includes an HIV construct. 

  • As outline above HIV tat at a 100ng/ml concentration did not have any significant cellular toxicity in  both the HTHU and the HTHU/HIV cell lines used in the study. The cART  (300 µM TFV + 50 µM FTC + 188nM DTG) concentration used in-vitro did not show any significant cell toxicity and were used  for several invitro studies by other investigators

References:

  • Perez-Valero I, Llibre JM, Castagna A, Pulido F, Molina JM, Esser S, Margot N, Shao Y, Temme L, Piontkowsky D, McNicholl IR, Haubrich R. Switching to Elvitegravir/Cobicistat/Emtricitabine /Tenofovir Alafenamide in Adults With HIV and M184V/I Mutation.J Acquir Immune Defic Syndr. 2021 Apr 1;86(4):490-495.
  • Margot N, Ram R, Abram M, Haubrich R, Callebaut C. Antiviral Activity of Tenofovir Alafenamide against HIV-1 with Thymidine Analog-Associated Mutations and M184V. Antimicrob Agents Chemother. 2020;64(4):e02557-19.
  • Callebaut C, Stepan G, Tian Y, Miller MD. In Vitro Virology Profile of Tenofovir Alafenamide, a Novel Oral Prodrug of Tenofovir with Improved Antiviral Activity Compared to That of Tenofovir Disoproxil Fumarate. Antimicrob Agents Chemother. 2015 Oct;59(10):5909-16.

Was there a concomitant increase in the mRNA for TRF1 and IF1B?  For consistency with the rest of the data in the study, assaying mRNA via qRT-PCR would be informative.

  • As suggested by the reviewer, we did additional QPCR experiments and quantitated mRNA expression of TRF-1 and IL-1β. These results are now included in the revised manuscript and are outlined in Figure 4C.

Minor Points:

  1. Line 93:  Telomere should be lower case (telomere).  It is also randomly capitalized in other places throughout the manuscript – as are other words as well.  Please check the manuscript carefully in revision.
  • We have now corrected this throughout the manuscript.
  1. Line 161: The Livak reference should be numbered as per journal style
  • We apologize for this omission. This is now corrected.
  1. Lines 245-246: change: A p value  was < 0.05 was  to:  A p value of < 0.05 was
  • This is now corrected.
  1. Lines 253-254: fix the wording in this run on sentence for clarity:  which were previously obtained and were stored at – 800C were used to quantitate TL and…
  • This is now corrected.
  1. Line 262: statistically is spelled incorrectly here and in other figure legends
  • This is now corrected.
  1. I’m intrigued by the tight error bars using these highly variable biological samples.  Just double checking that the standard deviations were calculated using biological and not technical replicates.
  • Standard deviations are calculated using biological replicates.
  1. Section 3.2: measuring mtDNA levels by QPCR is not really a measure of ‘mitochondrial DNA expression” (aka transcription) per se.  I would recommend rewording for clarity
  • This is now corrected.
  1. Line 282: telomere length is correct – not ‘telomere length expression’
  • This is now corrected.
  1. As implied by the above points, a carefully editing of the manuscript for spelling, capitalization and grammatical issues would be recommended.
  • We have made all specific changes as suggested and have also reviewed the manuscript for language, grammatical and editing errors and have corrected those.

We have made every attempt to address all the concerns raised by the reviewers and have revised the manuscript substantially. Changes to the revised manuscript are highlighted in yellow. We thank the reviewers for their comments and hope that our revised manuscript is now acceptable for publication.

Round 2

Reviewer 2 Report

The revised manuscript is improved.  However to support the data regarding the extraordinarily fast telomere shortening that is observed within 24 hrs. that was pointed out as a major concern in the initial review, the authors will need to actually measure the replication rate of their cells rather than simply speculate on it.  The data are hard to accept without the cell division data.

Author Response

Please find attached  data from additional experiments done as requested.

Round 3

Reviewer 2 Report

The new data provided by the authors supports their conclusions.  I find the manuscript to be improved and more convincing.